# Prediction of Carbonation Progress in Concrete Containing Calcareous Fly Ash Co-Binder

**DOI:** 10.3390/ma12172665

**Published:** 2019-08-21

**Authors:** Piotr Woyciechowski, Paweł Woliński, Grzegorz Adamczewski

**Affiliations:** 1Department of Building Materials Engineering, Warsaw University of Technology, 00-637 Warszawa, Poland; 2Faculty of Applied Sciences, Collegium Mazovia Innovative School, 08-110 Siedlce, Poland

**Keywords:** concrete, durability, carbonation modeling, calcareous fly ash

## Abstract

According to the European Standards (EN 450-1, EN 206), it is not permissible to use calcareous fly ash as an additive to concrete. However, other standards (for example, the American and Canadian ones) allow the use of high-calcium fly ash (type C) in concrete. As a result of brown coal combustion, a large amount of this type of fly ash is produced, and considerations on their use in concrete are in progress. Research into the influence of high-calcium fly ash on concrete durability is fundamental for dealing with that issue. The aim of the present research was to develop a new model of carbonation over time, also including calcareous fly ash content in the binder. The self-terminating model of carbonation is new, and not developed by other authors. In the current research, the former simplest model (a function of w/c ratio and time) is expanded with the calcareous fly ash to cement ratio. The basis is a statistically planned experiment with a large scope of two material variables (w/c ratio and fly ash to cement ratio). The main measured property is the carbonation depth after exposure to 4% of CO_2_ concentration (according to CEN/TS 12390-12). The model of carbonation obtained from this experiment is an output of the paper. Also, the idea of developing similar models for concrete families as a tool for designing concrete cover thickness for reinforced elements is described in the paper.

## 1. Introduction

A common cause of damage to concrete structures with steel reinforcement is the corrosion of steel resulting from insufficient protection. The protective abilities of concrete decrease due to the effect of physical and chemical factors over time. An important factor is carbonation, which decreases the concrete’s pH value. The approach of using the carbonation model elaborated in the research (on the basis of the collected data) and statistic curve-fitting for the results obtained for the tested concrete, enables us to design the concrete cover thickness for individual cases on the basis of the actual protective abilities of the concrete used.

The development of universal carbonation models for various types of concrete can lead to the creation of useful tools for designing durable structures in XC (carbonation threat) classes of exposure according to Eurocode EC2.

The course of carbonation in typical atmospheric conditions depends primarily on the material characteristics of the concrete, including the w/c ratio and the qualitative as well as quantitative composition of the binder. Most of the test results confirm that the use of fly ash in appropriate proportions not only impairs the protective properties of concrete against reinforcing steel [1,2], but also increases the concrete tightness [3]. Especially when chlorides from seawater or deicing salts, etc. cause a threat, the introduction of ash to cement or concrete can be very beneficial. However, in a heavily contaminated environment with high concentrations of carbon dioxide and chlorides, fly ash used as a partial substitution for the cement can accelerate the corrosion process of the reinforcement. For these reasons, the use of fly ash in concrete technology should be approached with great caution and preceded by a series of tests confirming the correctness of the solution [1].

The resistance to carbonation of concretes containing fly ash depends on its microstructure. Fly ash partially binds Ca(OH)_2_ due to the pozzolanic reaction. It leads to a decrease in the content of Ca(OH)_2_ in concrete. Due to the carbonation depth being higher, the same amount of CO_2_ could carbonate a larger volume of concrete, as Neville states [4] on the basis of Bier’s research [5]. Bier’s study proves that the carbonation rate is higher when the quantity of Ca(OH)_2_ in the cement paste is lower. In such a way, the fly ash presence in concrete may accelerate the rate of carbonation and increase the carbonated zone in concrete. However, the reverse effect connected with the formation of a denser, hardened paste with fly ash could also be observed. This leads to a reduction in the diffusivity and rate of carbonation limitation. Therefore, it could be concluded that the influence of fly ash on carbonation includes two contradictory effects:accelerating—connected with the shortage of Ca(OH)_2_ used for the pozzolanic reaction, which effects a deeper diffusion of carbon dioxide due to the concentration gradient;inhibitory—connected with the denser microstructure of the paste thanks to the physical effect of fine grains of ash and the chemical effect of the products of the pozzolanic reaction.

The subject of the research presented in this article is concretes made with fly ash, classified as calcareous according to PN-EN 197-1 [6] and according to the USA [7] and Canadian [8] standards, due to the high content of reactive lime, i.e., 15% by mass. The terms “calcareous fly ash” and “high-calcium fly ash” are used interchangeably in the literature [9,10,11], while from a formal point of view this nomenclature is regulated internationally by various standards, in which different criteria for ash differentiation are applied (Table 1).

There are different requirements, according to PN-EN 197-1 [6], for fly ash used as a main component of ordinary cements and ash used as a type II additive for concrete according to PN-EN 206 [12]. A reactive CaO content above 10% by mass formally excludes their use as a substitute for cements in concrete in EU countries, in the light of the requirements of PN-EN 450-1 [13]. Standard regulations in the USA or Canada [7,8], however, do not limit the use of high-CaO fly ash. In EU countries, the use of fly ash as a component of cement is not limited by the content of CaO. According to PN-EN 197-1, two types of fly ash are distinguished in this respect, siliceous and calcareous, and the limit of CaO is 10%.

In this article the term “calcareous fly ash” is used, defining it as fly ash for the production of cement in accordance with PN-EN 197-1, but using it as a component of concrete, despite not meeting the requirements of PN-EN 450.

It is worth noting that the results of published research are difficult to compare directly due to the different types of ash used, their content in relation to the cement mass, different concrete formulas and curing conditions, and different methods of conducting the experiments. There is a lot of research testing the influence of siliceous fly ash on concrete properties, among them carbonation: for example, the studies published by Kurda at al. (2019) [14], Ghorbani at al. (2019) [15], Carevic (2019) [16], Hussain at al. (2017) [17], Cai-feng Lu at al. (2018) [18], Branch at al. (2018) [19], Ying Chen at al. (2018) [20], and many others. However, during the last few years there have been only a few publications concerning the carbonation of concrete containing high-calcium fly ash as an additive [21,22,23,24,25,26].

The results of the published studies do not agree [21,22] as to which effect (accelerating or inhibitory) is dominant in the case of the carbonation of concrete with calcareous ash [27]. It depends on the interaction of compounds and external agents; however, one of the most important factors is the curing regime. Proper curing of concrete is crucial for pozzolanic reactions and advantageous for achieving the microstructure densifying effect. It was found that fly ash concrete that was not cured in the first days after concreting could very rapidly carbonate—even a 20 mm depth of carbonation was observed after a one-year exposure in the urban atmosphere [27]. 

First of all, the method of introducing ash into the concrete mix is important, i.e., whether the additive is introduced as a substitute of part of the cement or as an increase in the amount of binder. In the first case, the effect of the Ca(OH)_2_ deficit strongly influences the carbonation progress; in the second case the role of the densifying effect is most important [28].

There are only a few publications on the research into carbonation of concretes with calcareous ash. The high content of free lime in the ash suggests that the availability of Ca(OH)_2_ for the carbonation reaction will be high at the beginning of the process. It is worth noting that the free lime in the ash is highly reactive [23,24,29,30]. We would expect, then, a high initial rate of carbonation. On the other hand, the intensity of the pozzolanic reaction will reduce the calcium hydroxide amount, while at the same time filling pores with reaction products. Thus, a reduction of CO_2_ diffusion capacity occurs, which inhibits the rate of carbonation. The research results presented in [7,31] indicate that the replacement of a part of the cement with calcareous fly ash in an amount of 10–20% does not significantly affect the course of electrochemical processes on the surface of reinforcing steel, without deteriorating the protective properties of concrete against reinforcement. At the same time, there are also published studies indicating an increase in the dynamics of the progress of carbonation of concrete with calcareous fly ash, especially when it has a high content in relation to cement [26,30]. The research work carried out by Wolinski et al. (2018) [30] also showed that the calcareous ash used as a substitute for a part of the aggregate (corresponding to 20‒50% of the cement mass) allowed them to obtain concrete with a very low carbonation depth (less than 10 mm after 70 days in 4% CO_2_ concentration). These studies indicated that, with a constant cement content and constant w/c ratio, there is an optimal ash content, leading to the lowest susceptibility to carbonation. The dynamics of carbonation depth development change when the ash content increases, in such a way that intensive progress is noticed after the 56th day in 4% of CO_2_ if the ash content is high.

Research on the development of universal models of carbonation and different material and technological variables has been conducted by many authors [3,32,33,34,35,36,37,38,39,40,41,42,43,44]. When searching for a model of carbonation, one difficult issue is describing the intensity of the carbon dioxide flow in concrete. The first Fick’s law, which is used to describe diffusion, assumes the constant microstructural characteristics of concrete over the time. In this way, the following carbonation model could be developed, in the form of a power function of carbonation depth over time:(1)$$x=\sqrt{\frac{2D\varphi_{ext}}{a}}\times \sqrt{t}$$
where: *x*—depth of carbonation; *D*—diffusion coefficient; *φ_ext_*—external concentration of CO_2_; *t*—time of carbonation; *a*—coefficient determining the amount of CO_2_ bound in the way of carbonation by unit volume of concrete in kg/m^3^, calculated according to the CEB Bulletin 238 [45] as: *a* = 0.75*∙C∙[CaO]∙αH∙(MCO*_2_*/MCaO)* (*C*—content of cement in concrete, kg/m^3^; *[CaO]*—CaO content in the cement composition; *αH*—degree of hydration of cement; *MCO*_2_*, MCaO*—molar masses). 

In practice, the most widely used model is simplified. It relates to an average constant relative humidity RH and carbon dioxide concentration in the environment and can be expressed in the form:(2)$$x=A\times t^{1/2\;}+B$$
where *A* is a constant depending on the diffusion coefficient, the ability of concrete to bind CO_2_ and CO_2_ concentration in the air, and *B* is an empirical factor accounting for the initiation period of carbonation. This model is used by most researchers [3,31,32,33,34,35,36,37,38,39,40,41,42,43,44], but it treats the carbonation as a process unlimited in concrete space and unlimited in time. The abovementioned models based on the first Fick’s law assume that concrete in which diffusion takes place will not change its microstructure over time. The assumption of a constant diffusion flux in Equation (1) in such a circumstances is not valid. This significant simplification of the description of the carbonation process does not take into account additional factors that lead to changes in diffusivity. The effect of the saturation of the pores with carbonation products makes it impossible to describe the process using Fick’s law. Carbonation decreases porosity, in particular capillary porosity, and reduces the permeability of the concrete. This self-terminating nature of the phenomenon was mentioned by Bakker [46], Hergenröder [47], Nilsson [48], and Fagerlund [49]. The idea was further developed by Czarnecki, Woyciechowski at al. [50,51,52,53,54,55]. According to their findings, concrete carbonation can be described with a hyperbolic function of its depth over time (reciprocal square root of time), which has an asymptotic value parallel to the time axis. This asymptote is the limit of carbonation depth. The traditional and hyperbolic models of carbonation are shown in Figure 1.

The hyperbolic carbonation model is expressed in the general formula:(3)$$h=f(t^{-\;0.5})$$

All results published in the literature [50,51,52,53,54,55] are well fitted to the hyperbolic model expressed in Equation (3), enriched with w/c ratio and early curing time factors:(4)$$h=a\left(w/c\right)+b\left(cp\right)+c(t^{-0.5})$$
where: *h*—depth of carbonation, mm, *w/c*—water-cement ratio, *cp*—early curing with water period, days, *t—*time of exposure, days, *a, b, c*—coefficients describing relevance of influence of w/c ratio, early curing and exposure time on depth of carbonation. It was stated that parameters (*a, b, c*) mainly depend on the binder properties, the presence of mineral additives, and, especially, on the CO_2_ concentration. This type of model was elaborated for many types of concrete, particularly with Portland, slag, and siliceous fly ash cement. SEM analyses published in the literature [51,52,55] show a different density of concrete in carbonated and non-carbonated zones for all the tested binder compositions. The hyperbolic model allows us to calculate the maximum depth of carbonation (the limits of the hyperbolic model) and compare it with the reinforcement cover thickness in the analyzed element. This allows us to predict the risk of corrosion due to the carbonation and to calculate the time when the carbonation front will reach the reinforcement surface. This moment could be interpreted as the time of corrosion initiation.

The aim of the research presented in this article was to develop a model of carbonation of concrete with calcareous ash as a function of two basic variables in the composition of concrete: the water/cement ratio and the ratio of mass content of fly ash to cement. The implementation of this objective required the assumption of a preliminary general form of the model and then conducting an experiment in accordance with the principles of statistical planning, enabling the determination of a model with high conformity to the test results.

A model for predicting the long-term depth of carbonation in natural conditions on the basis of short-term tests in accelerated conditions was developed in the paper for concrete containing calcareous fly ash. The article also presents a simplified scheme of actions, enabling the use of a developed model for determining the optimal thickness of the reinforcement cover in exposure to carbonation.

## 2. Materials and Methods

We studied concretes with CEM I 32,5 R cement, river sand 0/2 mm (fineness modulus: 2.96), gravel aggregate 2/16 mm and fly ash, classified as calcareous according to PN-EN 197-1 [6], due to the high content of reactive lime: higher than 15% by mass. The fly ash used in the research was obtained from the combustion of brown coal in power plants. The grain size distribution was typical for this type of fly ash, according to analyses published in [24] and [56].

Basic physical (Table 2, Figure 2) and chemical (Table 3) properties allow us to categorize the fly ash as a component of cement according to PN-EN 197-1, for common cements. 

The study of the activity index (Table 4) according to PN-EN 450-1 [11] indicates that it is an active material. The literature [7,23,24] indicates that calcareous ashes are materials with both pozzolanic and latent hydraulic properties, and the test result is the combined effect of both properties.

As variables in the basic research program, w/c ratio (from 0.35 to 0.55) and ash to cement mass ratio (from 0.20 to 0.50) were assumed. The consistency was constant (12 ± 2 cm of slump) and adjusted with the help of a superplasticizer. The experimental program was developed on the basis of a two-factor, partial, quasi-uniform plan with a double repetition of the experiment at the central point (Table 5). The values x1 and x2 are the coded values of variables characteristic of the abovementioned experimental plan that lead to the best statistically representative answer with minimal testing [57]. The compositions of concrete mixtures, determined empirically to achieve the assumed consistency, are presented in Table 6. 

Concrete specimens for all tests were demolded after one day and then cured for 27 days in water, under laboratory conditions.

The progress of carbonation depth under accelerated conditions as well as the compressive strength after 28, 56, and 90 days and the tensile splitting strength after 28 days (i.e., at the start of accelerated carbonation exposure) were investigated. The compressive strength was determined on cubic specimens of 150 mm per side according to PN-EN 12390-3 [58], with the mean values determined from no fewer than five samples. The tensile strength for splitting was tested on cubic specimens of 150 mm per side according to PN-EN 12390-6 [59]. The depth of carbonation was tested according to the draft CEN TC 12390-12 [60]: CO_2_ concentration = 4%, *t* = 20 ± 2, RH = 50–60%, front measurement with phenolphthalein indicator, after exposure time of 14, 28, 56, 70, and 90 days. Beams 100 × 100 × 500 mm were used for the tests, with the two opposite long side surfaces of the specimen exposed to carbon dioxide. The remaining surfaces were covered with a paraffin coating just before placing the samples in the carbonation chamber. Three samples were tested for each composition. 

Scanning electron microscopy (SEM, TM3000, Hitachi, Japan) was used to characterize the microstructure of the chosen compositions of carbonized and noncarbonized two-year-old concrete specimens. 

## 3. Results

The strength characteristics of the tested concrete (Table 7) indicate that concretes with the addition of calcareous ash show an increase in strength between 28 and 90 days at 10–20% depending on the w/c ratio and the content of ash in the binder. The presence of ash in the binder at a constant w/c ratio caused a slight increase in the compressive and tensile strength. These results are in line with the literature [9,23,24,61].

The main goal of the research was to analyze the progress of carbonation of concrete over time and to attempt to mathematically describe this phenomenon in a practically useful way. The results of the depth measurement of carbonation with the phenolphthalein test after subsequent exposure periods in the carbonation chamber are summarized in Table 8. The given values were calculated as arithmetic averages of 10 measurements on each of the three specimens of the series (five on each of the two lateral surfaces of a single sample) [62].

It has been observed that, in the case of a low cement ratio, the presence of unreacted calcareous fly ash particles was more clearly observed (Figure 3 versus Figure 4 and Figure 5 versus Figure 6). After carbonation in concrete with a low w/c ratio (Figure 5), there are still many fly ash grains, which are not involved in the carbonation process. In the case of w/c = 0.52 after carbonation, fewer fly ash particles and portlandite crystals were visible due to the generally higher homogeneity of the microstructure. These observations are correlated with the higher positive effect of fly ash addition on the resistance of carbonation with a higher w/c ratio. It can be concluded that an optimal content of fly ash from the point of view of carbonation resistance exists and is dependent on the w/c ratio of concrete. This observation was discussed by Wolinski et al. (2015) [63].

## 4. Discussion

The results of carbonation depth were used to formulate a model expressing the depth of carbonation *h_t_* (after a predetermined time *t* under assumed exposure conditions) as a function of the water‒cement ratio (*w/c*) and the mass proportion of ash and cement in the binder (*p/c*), as follows:*h_t_*(*w/c*; *p/c*) = *a* + *b*(*w/c*) + *c*(*p/c*) + *d*(*w/c*)^2^ + *e*(*w/c*)(*p/c*) + *f*(*p/c*)^2^(5)

Response surfaces, including the total range of w/c ratio values from 0.35 to 0.55 and p/c values from 0.2 to 0.50 assumed in the tests, are presented in Figure 7, Figure 8 and Figure 9, corresponding to selected exposure times *t*, i.e., 56, 70, and 90 days, with the determination coefficient in all cases significantly exceeding the value of 0.9.

For a complete set of results, a general concrete carbonation model with fly ash was developed, taking into account both material variables (w/c ratio and p/c ratio) and the time of carbonation under specific exposure conditions. The general form of the model was assumed to take into account the finite nature of the carbonation process over time, consistent with the considerations presented in the introduction to this article. Finally, the following general form of the model was adopted:*h*(*w/c*;*p/c*;*t*) = *a* + *b*_1_ × (*w/c*) + *b*_2_ × (*p/c*) + *b*_3_/*sqrt*(*t*)(6)
where: *h*—depth of carbonation, mm; *w/c*—water-cement ratio; *p/c*—powder to cement ratio; *a, b*_1_*, b*_2_*, b_3_*—material and technological coefficients, *t*—time of exposure, days. 

The model does not take into account the essential technological factor, which is the time of early curing of concrete, because all tests were carried out with reference to samples hardening for up to 28 days in water. The possible practical use of models of the proposed type requires taking into account this factor in the form of an additional expression or limiting the validity of the equation to a specific regime of concrete curing. The detailed form of the proposed model determined by the curve fitting method, based on the experimental results, is as follows: *h*(*w*/*c*; *p/c*; *t*) = 1.07 + 25.28 × (*w/c*) – 3.53 × (*p/c*) – 41.07/*sqrt*(*t*)(7)
and is characterized by the determination coefficient 0.85. This indicates the good fit of the adopted model to the results of laboratory tests obtained under accelerated carbonation conditions in 4% CO_2_. The presented detailed model can be applied only to concrete with the range of constant and changed material and technological variables used in the research, but the method of its determination is an example that can be repeated in the case of various material and technological assumptions.

For further verification of model fitting to the experimental results, values calculated from the model and obtained from the experiments over 90 days of exposure in 4% of CO_2_ concentration were compared, as in Table 9. The accuracy of the prediction is ± 15%, which is an acceptable value from an engineering point of view.

Predicting the course of carbonation in a real structure using the developed model requires establishing an equation expressing the dependence of the progress of carbonation on the CO_2_ concentration in the research environment and in the exploitation environment. The dependence of carbonation depth on CO_2_ concentration is described in many publications [37,40,64], with a form similar to the following:(8)$$X_{1}=X_{2}\times \frac{c_{1}\times t_{1}}{c_{2}\times t_{2}}$$
where: *X*_1_—depth of carbonation after exploitation in c_1_ concentration after t_1_ time of exploitation; *X*_2_—depth of carbonation in accelerated conditions, *t*_2_—time of exposure in accelerated conditions; *c*_2_—CO_2_ concentration in accelerated conditions; *t*_1_—expected service life of construction; *c*_1_—CO_2_ concentration in exploitation conditions. Please check that intended meaning has been retained

Thus, having the developed detailed model in the general form (Equation (6)), i.e., a form that assumes the finite character of the carbonation process, we can calculate the limit of functions at *t* reaching to infinity, i.e., the ordinate of the model asymptote denotes the maximum possible depth of carbonation in the test conditions. In the analyzed case it means under accelerated conditions (CO_2_ concentrations equal to c_2_ = 4%). Assuming this value as *X*_2_ in Equation (8), and as *t*_2_ (the maximum test time under accelerated carbonation conditions), i.e., 90 days, one can determine the predicted carbonation depth after time *t*_1_ (e.g., assumed service life of 50 or 100 years) at a given concentration of carbon dioxide *c*_1_ (e.g., atmospheric concentration—approximately 0.04%). In Table 10 a simulation of the carbonation depth using the above dependence after 50 and 100 years of service life in the atmosphere with a concentration of 0.04% carbon dioxide is shown.

The values presented in Table 10 can be considered as the starting point for determining the required reinforcement cover in a construction made of concrete with given w/c and p/c ratios. These values should be increased by a safety factor, taking into account random factors and those related to the curing of concrete. This is an alternative approach to the standard procedure for determining the thickness of the reinforcement cover indicated by the EC2 standard [65], which is favorable due to the optimal method of determining the minimum safe thickness due to durability, taking into account the concrete material specification. The proposed procedure for determining the required cover is presented in the form of the diagram (Figure 10) for the concrete family. As a concrete family, a set of technologically similar concretes was defined, characterized by a constant type of cement, type of ash, chemical admixture, initial care method, and the following variables: any type of aggregate (except lightweight), any type of water (in accordance with PN-EN 1008 [66]), the specified range of variability of w/c and p/c ratios, and dosage of the same admixture in the amount necessary to the required consistency.

## 5. Conclusions

As a result of the research presented in this article, the following research objectives have been met:It was found that an increase in the content of calcium fly ash used as a partial cement substitute (a fly ash to cement ratio between 0.2 and 0.5) in concrete at a constant w/c ratio caused a slight increase in the compressive and tensile strength;A carbonation range model was developed as a function of the w/c and calcareous fly ash to cement ratio in the form of *h_ti_ = f (w/c, p/c)*, after different exposure times *t_i_*, i.e., after 56, 70, and 90 days of exposure;A model of the progress of carbonation in the form of *h = f(1/sqrt(t))* has been formulated and verified in relation to concretes with calcareous fly ash of various *w/*c and *p/c* ratios;A general model for concrete carbonation with calcareous fly ash *h = f(w/c; p/c; 1/(sqrt(t))* has been developed, taking into account all the variables used in the research;

It was shown that the developed general model of carbonation for concrete with a wide range of water/cement and ash to cement ratios and a constant set of components as well as fixed technological procedures could be a useful tool for the experimental design of optimal cover thickness.

## Figures and Tables

**Figure 1 materials-12-02665-f001:**
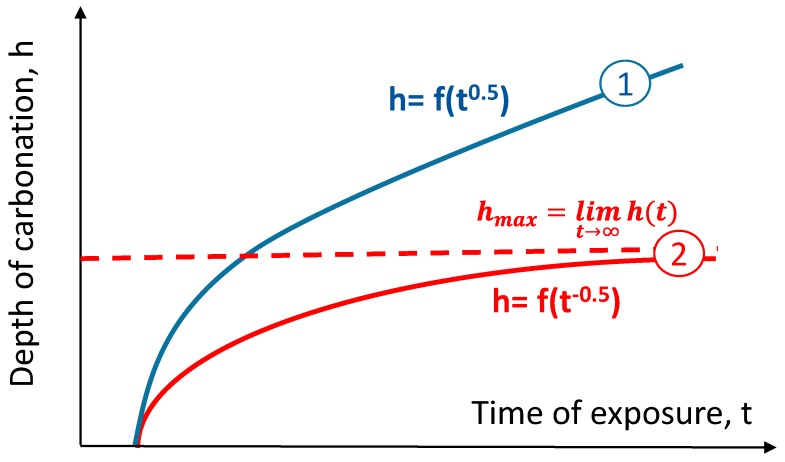
Traditional” power (1) and hyperbolic (2) models of carbonation phenomena.

**Figure 2 materials-12-02665-f002:**
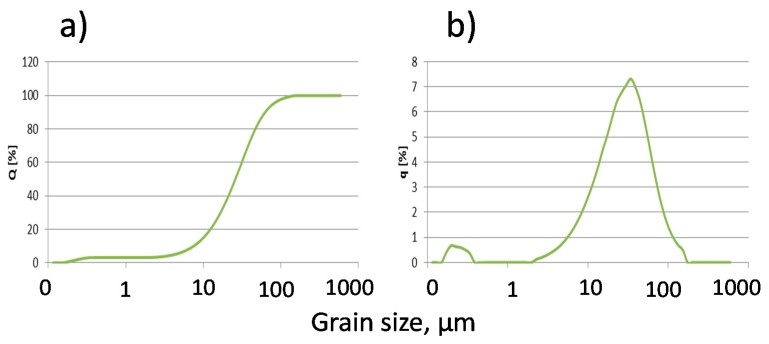
Calcareous fly ash grain size distribution (laser granulometry): (**a**): cumulative curve (**b**): population curve.

**Figure 3 materials-12-02665-f003:**
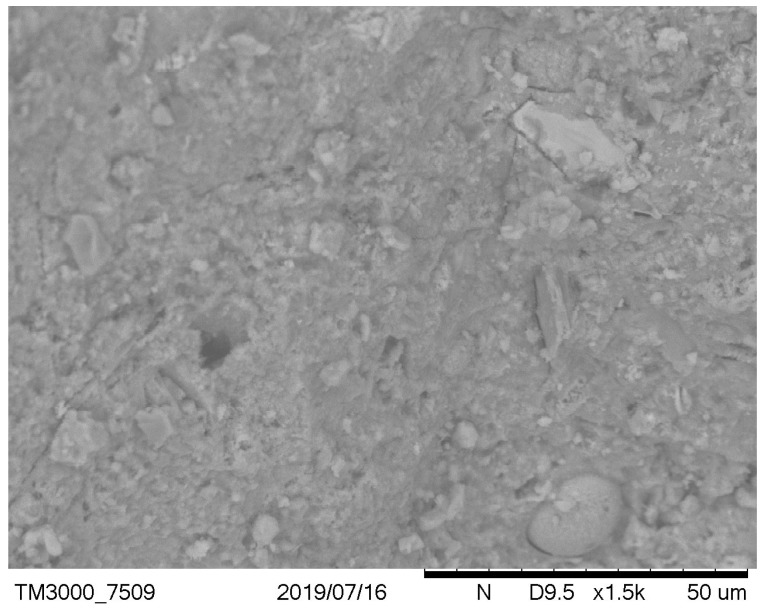
Noncarbonated concrete microstructure—mix symbol: 8, w/c = 0.38, p/c = 0.46.

**Figure 4 materials-12-02665-f004:**
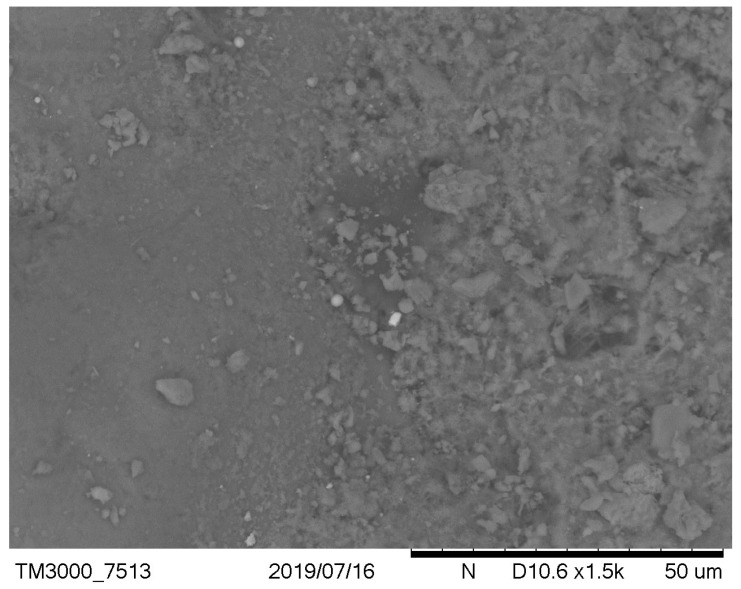
Noncarbonated concrete microstructure—mix symbol: 2, w/c = 0.52, p/c = 0.46.

**Figure 5 materials-12-02665-f005:**
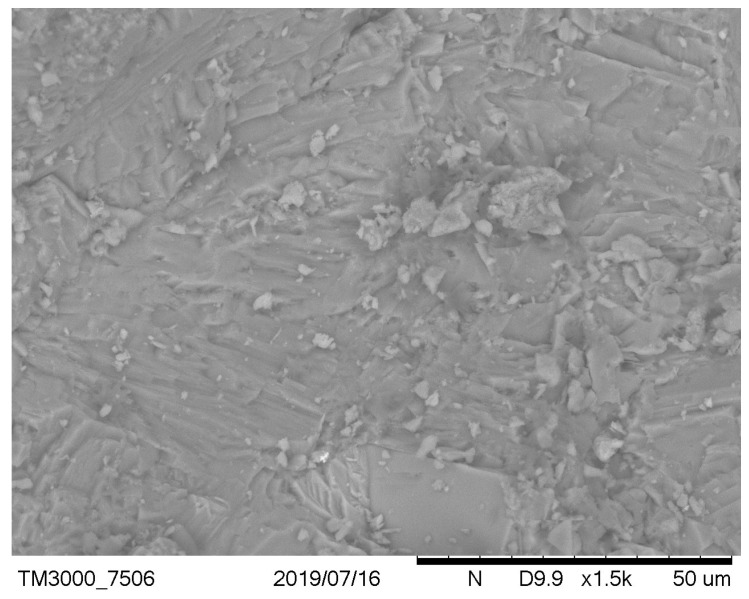
Carbonated concrete microstructure—mix symbol: 8, w/c = 0.38, p/c = 0.46.

**Figure 6 materials-12-02665-f006:**
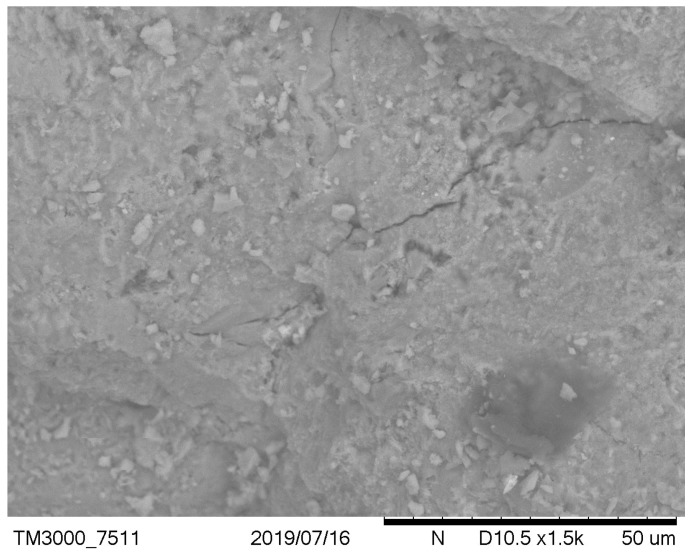
Carbonated concrete microstructure—mix symbol: 2 w/c = 0.52, p/c = 0.46.

**Figure 7 materials-12-02665-f007:**
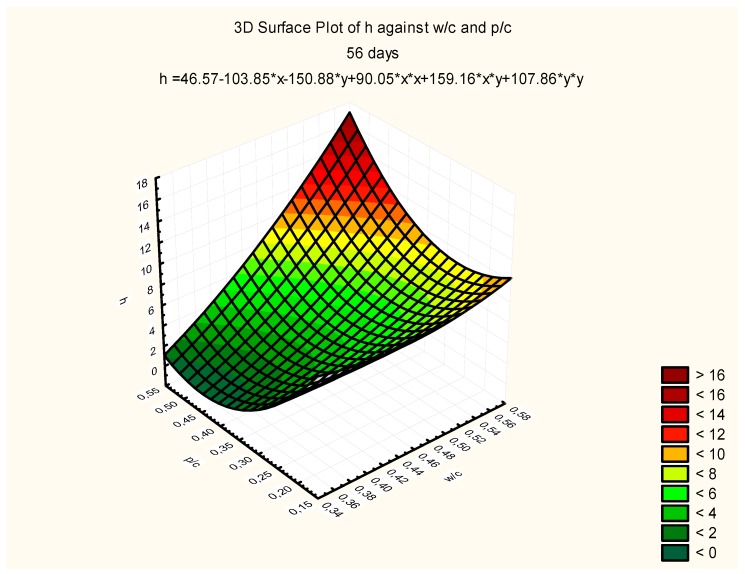
Depth of carbonation as a function of w/c and p/c, after exposure time *t* = 56 days, *R*^2^ = 0.93.

**Figure 8 materials-12-02665-f008:**
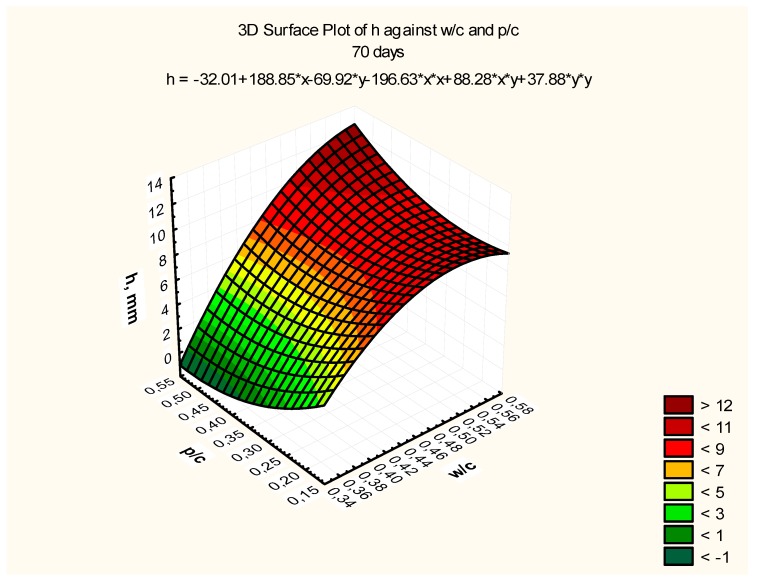
Depth of carbonation as a function of w/c and p/c, after exposure time *t* = 70 days, *R^2^* = 0.92.

**Figure 9 materials-12-02665-f009:**
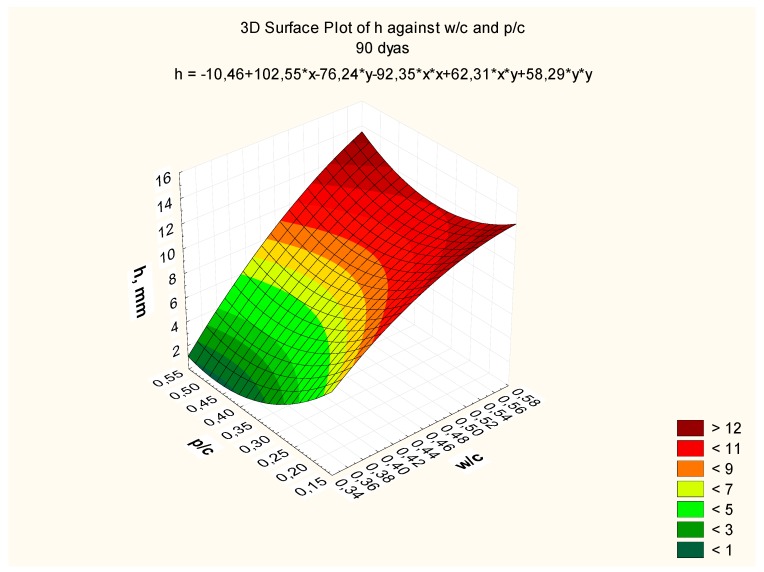
Depth of carbonation as a function of w/c and p/c, after exposure time *t* = 90 days, *R^2^* = 0.95.

**Figure 10 materials-12-02665-f010:**
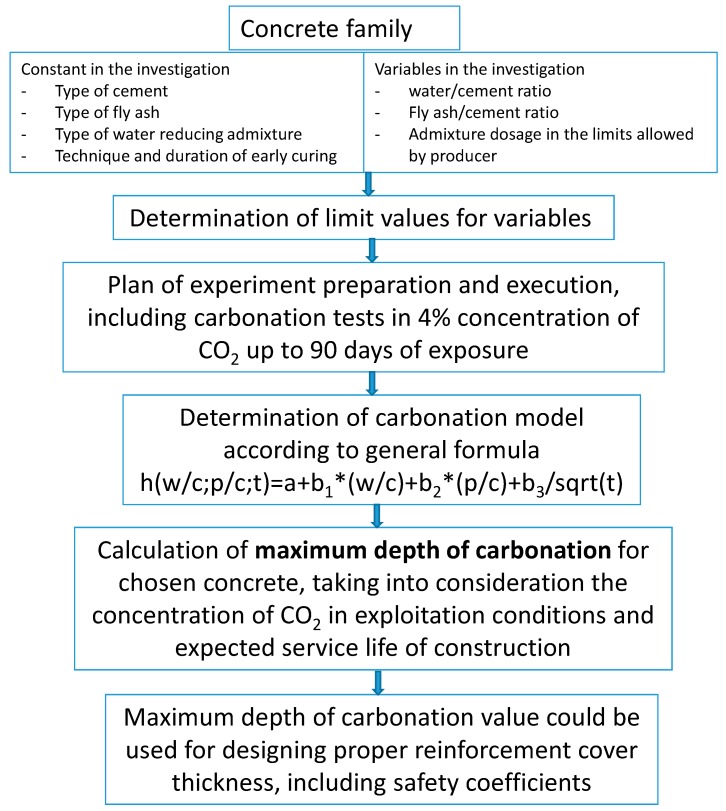
Proposed procedure for determining the thickness of the reinforcement cover with use of the developed model of carbonation.

**Table 1 materials-12-02665-t001:** Requirements for high-calcium (calcareous) fly ash according to different standards.

Requirements		ASTM 618	CAN/CSA A3000		PN EN 450-1		PN EN 197-1	
		Class C	Class CI	Class CH	Case 1	Case 2	Case 1	Case 2
total CaO content, %		≥10.0	8 ÷ 20	≥20.0	-	-	-	-
reactive CaO content, %		-	-	-	≤10	≤10	10 ÷ 15	>15
free CaO content, %		-	-	-	≤1.5	>1.5	-	-
volume stability, mm		-	-	-	-	≤10	≤10	≤10
reactive SiO_2_ content, %		-	-	-	-	-	≥25	-
SiO_2_ + Al_2_O_3_ + Fe_2_O_3_ content, %		≥ 50.0	-	-	≥70.0	≥70.0	-	-
SO_3_ content, %		≤5.0	≤5.0	≤5.0	≤3.0	≤3.0	≤3.0	-
Activity index, %	7 days	≥75.0	≥68.0	≥68.0	-	-	-	-
	28 days	-	-	-	≥75.0	≥75.0	-	-
	90 days	-	-	-	≥85.0	≥85.0	-	-
Water demand, %		≤105.0	-	-	≤95	≤95	-	-

**Table 2 materials-12-02665-t002:** Basic properties of calcareous fly ash used in research.

Property	Density, *ρ* g/cm^3^	Min Grain Size, D_min_ µm	Max Grain Size, D_max_ µm	Specific Area, cm^2^/cm^3^
Result	2.27	0.2	174.6	8677

**Table 3 materials-12-02665-t003:** Characteristics of the chemical composition of calcareous fly ash.

Lp.	Composition Characteristic	Mass Content, %
1.	SiO_2_	47.06
2.	Al_2_O_3_	18.40
3.	Reactive CaO	15.20
4.	SO_3_	1.94
5.	CO_2_	0.58
6.	Loss of ignition	3.60
7.	Other compounds	15.70

**Table 4 materials-12-02665-t004:** Activity index (according to PN-EN 450-1) for calcareous fly ash, f_c28_, f_c90_—compressive strength after 28 and 90 days; WA_28_, WA_90_—calculated as a compressive strength of reference mortar and fly ash mortar ratio in %.

Lp.	Type of Material		f_c28_, MPa	WA_28_, %	f_c90_, MPa	WA_90_, %
1	Reference mortar		41.4	83.0	49.5	85.9
2	Mortar with 25% of fly ash as cement substitution		49.9		57.6	

**Table 5 materials-12-02665-t005:** Range of variables used in experiment.

Composition Symbol	Coded Variables Values		Real Variables Values	
	x1	x2	x1’ (w/c)	x2’ (p/c)
1	−1	−1	0.379	0.24
2	1	1	0.521	0.46
3	−1.414	0	0.35	0.35
4	1.414	0	0.55	0.35
5	0	−1.414	0.45	0.20
6	0	1.414	0.45	0.50
7	0	0	0.45	0.35
8	−1	1	0.379	0.46
9	1	−1	0.521	0.24
10	0	0	0.45	0.35

**Table 6 materials-12-02665-t006:** Concrete mix compositions (constant value of cement and fly ash to aggregate mass coefficient = 0.24; constant consistence 12 ± 2 cm of slump), w/c—water/cement mass ratio, p/c—fly ash/cement mass ratio.

Composition Symbol	Components Content, kg/m^3^						Superplasticizer, % of Cement Mass
	Cement	Water	Aggregate	Fly Ash	w/c	p/c	
1	361.2	137	1872	88.1	0.379	0.244	2.5
2	299.6	156	1817	136.6	0.521	0.456	1
3	338.6	118.5	1904	118,5	0.35	0.35	2.2
4	317.3	174.5	1784	111.0	0.55	0.35	0.1
5	363.3	163.5	1816	72.6	0.45	0.2	0.4
6	297.8	134	1861	148.8	0.45	0.5	1.8
7	327.3	147.3	1841	114.5	0.45	0.35	1.44
8	312.4	118.5	1895	142.5	0.379	0.456	2.55
9	343.8	179	1781	83.5	0.521	0.244	0.48
10	327.3	147.3	1841	114.5	0.45	0.35	1.44

**Table 7 materials-12-02665-t007:** Compressive and tensile strength results.

Mix Composition Symbol and Characteristics	Average Compressive Strength (MPa)			Average Tensile Strength (MPa)
	Standard Deviation (MPa)			Standard Deviation (MPa)
	Concrete Age (days)			
	28	56	90	28
1 w/c = 0.379; p/c = 0.24	65.6	66.1	68.4	4.50
	0.67	2.20	0.80	0.07
2 w/c = 0.521; p/c = 0.46	54.4	57.6	58.7	3.40
	1.45	0.72	1.30	0.14
3 w/c = 0.35; p/c = 0.35	68.0	68.4	73.3	4.40
	0.65	2.92	1.76	0.04
4 w/c = 0.55; p/c = 0.35	43.2	45.5	46.6	2.90
	1.73	0.62	2.47	0.19
5 w/c = 0.45; p/c = 0.2	62.3	65.3	65.8	4.60
	0.86	1.11	0.46	0.06
6 w/c = 0.45; p/c = 0.5	61.4	67.9	73.4	3.80
	1.02	1.30	0.96	0.05
7 w/c = 0.45; p/c = 0.35	66.4	68.6	70.5	4.50
	2.25	1.32	2.26	0.10
8 w/c = 0.379; p/c = 0.46	72.0	73.0	78.8	5.20
	1.37	0.99	1.53	0.14
9 w/c = 0.521; p/c = 0.24	44.2	47.8	48.2	3.30
	1.01	0.54	1.98	0.28

**Table 8 materials-12-02665-t008:** Average values of measured depths of carbonation.

Exposure Time, Days	Mix Composition Symbol								
	1	2	3	4	5	6	7	8	9
	Average Depth of Carbonation (mm)								
14	1.47	0.86	0.23	0.99	1.04	1.06	1.32	0.25	0.85
28	4.68	2.03	0.54	3.44	2.20	2.89	1.57	1.17	2.32
56	5.05	8.48	0.63	7.82	6.31	5.16	3.58	1.76	6.81
70	5.70	8.58	0.60	10.42	7.80	8.89	7.19	2.02	9.51
90	6.22	8.29	2.45	11.63	9.25	9.30	7.25	2.09	10.44

**Table 9 materials-12-02665-t009:** Comparison of experimental and calculated values of depth of carbonation.

Mix Symbol	Average Depth of Carbonation after 90 Days in 4% of CO_2_		
	Calculated from Model (7), mm	Measured during Experiment, mm	Relative Difference, % *((c − b)/b) × 100%*
*(a)*	*(b)*	*(c)*	*(d)*
1	5.7	6.2	8
2	8.3	8.3	0
3	2.8	2.5	−14
4	9.9	11.6	15
5	7.9	9.3	15
6	8.0	9.3	14
7 and 10 (average)	7.9	8.8	10
8	2.4	2.1	−15
9	9.1	10.4	13

**Table 10 materials-12-02665-t010:** Calculated depths of carbonation (mm) after 50 and 100 years in a natural atmosphere.

w/c	p/c				
	0.2	0.24	0.35	0.46	0.5
	50 years				
0.35	19	20	18	17	16
0.379	20	20	19	18	18
0.45	24	24	23	22	22
0.521	27	27	26	26	25
0.55	29	29	28	27	27
	100 years				
0.35	37	39	35	34	33
0.379	40	39	38	37	36
0.45	48	47	45	44	43
0.521	54	54	53	51	50
0.55	58	57	56	54	53

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
