# Peer review of "Prediction of Carbonation Progress in Concrete Containing Calcareous Fly Ash Co-Binder"

_materials, 2019, doi:10.3390/ma12172665_

Round 1
Reviewer 1 Report
Abstract:
The abstract of this study is not clear. The authors mixed introduction, background, methodology. The aim, methods and the main outputs of this work should be presented in proper and systematic way. There is not any output of this study in the abstract.
Line 15. Replace “,” with “.”.
Line 15. Fly Ash to cement ratio (p/c). “P” stands for what?
Introduction:
The literature review of this study made almost four years ago (see, the list of bibliography). For example, they were cited only few studies from 2017-2019. In fact, they only showed few similar studies (please, see other studies how they prove the novelty of their work, e.g., Rawaz Kurda). Therefore, the whole paper needs to be adjusted for the current time (23/07/2019), namely the background of this study. The authors were failed to prove the novelty of this study. The authors cannot prove the novelty of their work without the recent literature.
In introduction. Define the main difference between calcareous fly ash and other types of fly ash.
Materials and Methods:
Figure 2. what is the difference between right and left figures?
Table 3 is misplaced. It should be after lines 198-202.
Table 3. Define WA.
Table 4. How X1 and X2 are defined? In other words, how did you get the values. Explain it in the manuscript.
Results:
The reviewer still cannot understand how the authors do the prediction of carbonation progress in concrete containing calcareous fly ash when water to cement ratio, cement content, superplasticizer content, water content are varied. The authors tried to explain the issue (e.g. Figuer 10), but it is still not clear.
Table 6. change “,” to “.”. Consider this comment in other cases.
3D figures. The reviewer suggests to add the original value (see, studies Rawaz Kurda) on the figure to see the reliability of the curve. Please, consider this comment in other figures.
Conclusion
First point is not clear. Do you mean for certain incorporation ratio of fly ash?
Author Response
Dear Respectable Reviewer
Thank you for your valuable and helpful comments and recommendations. All of them were considered and all improvements were done. Above there are answers for your comments and info how the comment was included in the paper.
Sincerely
Corresponding author
Answers for the Reviewer 1
The abstract of this study is not clear. The authors mixed introduction, background, methodology. The aim, methods and the main outputs of this work should be presented in proper and systematic way. There is not any output of this study in the abstract.
The abstract is rewritten with use of Reviewer recommendation
Line 15. Replace “,” with “.”.
In all numbers (in text and in tables) symbol “,” was replaced with “.”
Line 15. Fly Ash to cement ratio (p/c). “P” stands for what?
Symbol “p” is taken from the word “powder” - the explanation of meaning of p/c symbol was added in the text
Introduction:
The literature review of this study made almost four years ago (see, the list of bibliography). For example, they were cited only few studies from 2017-2019. In fact, they only showed few similar studies (please, see other studies how they prove the novelty of their work, e.g., Rawaz Kurda). Therefore, the whole paper needs to be adjusted for the current time (23/07/2019), namely the background of this study. The authors were failed to prove the novelty of this study. The authors cannot prove the novelty of their work without the recent literature.
Few newer publications were added to the references, among them interesting and useful publications of Kurad - recommended by the Reviewer. The problem of high-calcium fly ash is not widely discussed in the recent publications on cementitious concrete. This material is considered as a compound of geopolymer, and we are now also working on this subject but that a different issue - so the set of publication which concerns this matter are not cited in the paper.
Also the issue of self-terminated model of carbonation is not developed by other Authors than my scientific team. Discussion of basic formula of this model was done in previous publications of mine, cited in the text. In this publication we would like to develop the additional element of the formula, connected with high-calcium fly ash content in binder.
In introduction. Define the main difference between calcareous fly ash and other types of fly ash.
The comparison is done, additionally the Table is added (Table 1) with comparison of different standard recommendations (and definitios) for high-calcium fly ash.
Materials and Methods:
Figure 2. what is the difference between right and left figures?
The difference is described - this is the formula of material granulometry recived from laser granulometer Horiba )frequency and population plots of measured data.
Table 3 is misplaced. It should be after lines 198-202.
Correction was done
Table 3. Define WA.
Definition is added in the table caption
Table 4. How X1 and X2 are defined? In other words, how did you get the values. Explain it in the manuscript.
Explanation is added on the basis of fundamental literature on statistical experiment design. Values x1 and x2 are codded values determined by adopted plan of experiment.
Results:
The reviewer still cannot understand how the authors do the prediction of carbonation progress in concrete containing calcareous fly ash when water to cement ratio, cement content, superplasticizer content, water content are varied. The authors tried to explain the issue (e.g. Figuer 10), but it is still not clear.
The analysis is done on the basis of statistical material modeling. Two variables are used w/c and fly ash/cement ratios and the mix proportions were calculated with constant fly ash+cement/aggregate ratio = 0,24, and superplasticizer dosage to obtain constant consistence 12 cm of slump. This method of experimental material model developing is statistically representative if determination coefficient is high. In presented research the value of R2 for developed models was 0,85 and higher which indicates the adequacy of the model to the results.
Table 6. change “,” to “.”. Consider this comment in other cases.
In all numbers (in text and in tables) symbol “,” was replaced with “.”
3D figures. The reviewer suggests to add the original value (see, studies Rawaz Kurda) on the figure to see the reliability of the curve. Please, consider this comment in other figures.
The reliability was discussed for 90days results in added table 9, the reliability of curves is proved by R2 values - this values are added in al 3D plots
Conclusion
First point is not clear. Do you mean for certain incorporation ratio of fly ash?
Correction is done to clear the authors intentions
Thank You for your help to improve the quality of our paper
Authors
Reviewer 2 Report
Reviewer's Comments
This paper presents an experimental study to develop an analytical model which permits to evaluate the carbonation depth in concrete containing calcareous fly ash varying water to cement ratio (W/c) from 0.35 to 0.55 and the fly ash to cement ratio (p/c) from 0.2 to 0.50. The results showed that the calcium fly ash used as a partial cement replacement in concrete produced a slight increase in compressive and tensile strength. The developed model to predict the carbonation depth in concrete made with fly ash is validated by experimental tests.
Some suggestions to enhance the manuscript quality are as follows:
1- The abstract should present the main conclusion found in this study.
2- The abbreviation should be defined for its first utilization in the text such as: EC (page 1, line 34); fc28, fc90, WA28, WA90 (Table 3); p/c, w/c (Table 5); SEM (page 4, line 163); acc. (page3, line118).
3- Page 2, line 87-88: The sentence in these lines is not clear and should be reformulated.
4- Page 3, line 96: The sentence in this line should be corrected as “Research results, presented, among others in literatures [7,22]”.
The same remark in line 101: “The research work carried out by Wolinski et al. (2018) [19] showed also that….”.
5- Page 5, Section 2: The fineness modulus of river sand should be presented in the text.
In line 188, the following sentence should be corrected as “for such a type of fly ash, according to analysis work published in literatures [21] and [49]”.
6- Replace the comma in the fraction number by a point in Tables 1 to 8 and also in the text.
7- Page 7, line 217: correct the sentence in this line as “after 28, 56 and 90 days…….”.
Also in line 235: ” These results are in line with literature [9,20, 21, 53]”
In Table 6, where are the tensile strength values for 56 and 90 days? Also, where are the results obtained for the concrete reference?
8- Page 8, line 257: replace the sentence in this line by “…….discussed by Wolinski et al. (2015) [55]”.
9-In figure 7, the axis titles and numbers should be defined.
10- The results of carbonation depth predicted from Eq. 7 should be compared with those obtained from the experimental tests. The compared results in terms of the carbonation depth could be presented in table to well understand the model precision.
11- In Table 8, the unit of the carbonation depth should be precised.
12- Page 14: In figure 10, the third paragraph should be corrected as” including carbonation tests in 4% concentration of CO2”

Author Response
Dear Respectable Reviewer
Thank you for your valuable and helpful comments and recommendations. All of them were considered and all improvements were done. Above there are answers for your comments and info how the comment was included in the paper.
Sincerely
Corresponding author
Answers for comments of Reviewer 2
This paper presents an experimental study to develop an analytical model which permits to evaluate the carbonation depth in concrete containing calcareous fly ash varying water to cement ratio (W/c) from 0.35 to 0.55 and the fly ash to cement ratio (p/c) from 0.2 to 0.50. The results showed that the calcium fly ash used as a partial cement replacement in concrete produced a slight increase in compressive and tensile strength. The developed model to predict the carbonation depth in concrete made with fly ash is validated by experimental tests.
Some suggestions to enhance the manuscript quality are as follows:
1- The abstract should present the main conclusion found in this study.
Abstract is rewritten
2- The abbreviation should be defined for its first utilization in the text such as: EC (page 1, line 34); fc28, fc90, WA28, WA90 (Table 3); p/c, w/c (Table 5); SEM (page 4, line 163); acc. (page3, line118).
Abbrewiations are defined
3- Page 2, line 87-88: The sentence in these lines is not clear and should be reformulated.
Sentence was reformulated
4- Page 3, line 96: The sentence in this line should be corrected as “Research results, presented, among others in literatures [7,22]”.
The same remark in line 101: “The research work carried out by Wolinski et al. (2018) [19] showed also that….”.
Sentence was reformulated
5- Page 5, Section 2: The fineness modulus of river sand should be presented in the text.
Fineness modulus value is added
In line 188, the following sentence should be corrected as “for such a type of fly ash, according to analysis work published in literatures [21] and [49]”.
Sentence wass reformulated
6- Replace the comma in the fraction number by a point in Tables 1 to 8 and also in the text.
In all numbers (in text and in tables) symbol “,” was replaced with “.”
7- Page 7, line 217: correct the sentence in this line as “after 28, 56 and 90 days…….”.
Also in line 235: ” These results are in line with literature [9,20, 21, 53]”
Sentence was reformulated
In Table 6, where are the tensile strength values for 56 and 90 days? Also, where are the results obtained for the concrete reference?
Tensile strength was tested only after 28 days. The reference concrete is not useful for material model development due to the not only quantitative but also qualitative change of the material which contains 0% of fly ash.
8- Page 8, line 257: replace the sentence in this line by “…….discussed by Wolinski et al. (2015) [55]”.
Sentence was reformulated
9-In figure 7, the axis titles and numbers should be defined.
Figure was corrected
10- The results of carbonation depth predicted from Eq. 7 should be compared with those obtained from the experimental tests. The compared results in terms of the carbonation depth could be presented in table to well understand the model precision.
The results of carbonation testing after 90 days of exposure shown in newly added table 9 were used for comparison of results obtained from experiment and calculated from the model. The precision of prediction carbonation depth is ± 15%, which is typical for alcacimetry and seems to be adequate for engineering purposes
11- In Table 8, the unit of the carbonation depth should be precised.
Units were added in the table
12- Page 14: In figure 10, the third paragraph should be corrected as” including carbonation tests in 4% concentration of CO2”
figure 10 was corrected
Thank you for your helpful advices which allowd us to impove the quality of our paper
Authors
Round 2
Reviewer 1 Report
Now that the manuscript has been revised according to the reviewer comments, it can be recommended for publication.